# Cellular Nickel-Yttria/Zirconia (Ni–YSZ) Cermet Foams: Manufacturing, Microstructure and Properties

**DOI:** 10.3390/ma13112437

**Published:** 2020-05-26

**Authors:** Ulf Betke, Katja Schelm, Andreas Rodak, Michael Scheffler

**Affiliations:** Institute for Materials and Joining Technology-Nonmetallic Inorganic Materials and Composites, Otto-von-Guericke-University Magdeburg, Große Steinernetischstraße 6, 39104 Magdeburg, Germany; katja.schelm@ovgu.de (K.S.); andreas.rodak@st.ovgu.de (A.R.); m.scheffler@ovgu.de (M.S.)

**Keywords:** cellular material, ceramic foam, ceramic metal composite (cermet), nickel-yttria-stabilized zirconia, Ni-ZrO_2_-Y_2_O_3_ (Ni–YSZ), chemical reduction, electrical conductivity

## Abstract

Open-celled ceramic composite foams were prepared from NiO and yttria-stabilized zirconia (YSZ) powders by the polymer sponge replication (Schwartzwalder) technique using the respective aqueous dispersions. Mechanically stable NiO–YSZ foams with an average porosity of 93 vol.% were obtained. After chemical reduction of the NiO phase with hydrogen, cellular Ni–YSZ cermet structures were obtained. They are characterized by an electric conductivity up to 19∙10^3^ S∙m^−1^ which can be adjusted by both, the Ni volume fraction, and the sintering/reduction procedure. The NiO–YSZ ceramic foams, as well as the cellular Ni–YSZ cermets prepared therefrom, were characterized with respect to their microstructure by scanning electron microscopy, confocal Raman microscopy and X-ray diffraction with Rietveld analysis. In addition, the compressive strength, the electric conductivity and the thermal conductivity were determined. The collected data were then correlated to the sample microstructure and porosity and were also applied for modelling of the mechanical and electric properties of the bulk Ni–YSZ strut material.

## 1. Introduction

Cellular structures are described for all of the material classes: metals, ceramics/glasses and polymers [1,2]. However, the knowledge of cellular composite materials is mostly limited to coated foam structures [3,4]. Cellular composites consisting of an interpenetrating phase network within the strut material are seldom described. Some examples are silicon carbide ceramic foams strut-infiltrated with liquid silicon (SiSiC foams, reference [5]), or oxide-bonded SiC foams functionalized with a metal phase by a salt infiltration and subsequent chemical reduction (MESCAL process, reference [6]). The resulting composite foams possess new functionalities—for example electric conductivity for the cellular metal@SiC structures or a good thermal stability for the SiSiC foams. Nevertheless, due to the separation of manufacturing the cellular structure from the following functionalization, the preparative effort is very high. Therefore, a different approach has been investigated within this study, which is well-known for the manufacturing of bulk ceramic–metal composites (cermets). A common cermet is Ni-ZrO_2_-Y_2_O_3_ (Ni–YSZ), which gained popularity as anode material in solid oxide fuel cells (SOFCs) due to its unique combination of ionic conductivity through the YSZ phase and electronic conductivity through the Ni phase [7]. Ni–YSZ composites for SOFCs are usually prepared by dispersion-based processes using a mixture of NiO and YSZ. After shaping and drying, the NiO is chemically reduced to metallic Ni. This is accompanied by the formation of a pore network within the cermet due to the density difference between NiO and Ni. This porosity is essential for the application of Ni–YSZ as anode material in SOFCs operated with H_2_ gas, respectively.

A common manufacturing process for cellular ceramic structures is the polymer sponge replication method, or Schwartzwalder technique [8]. This process is based on coating a ceramic dispersion onto a polymer foam template and a subsequent thermal removal of the polymer structure. The resulting green body is then sintered forming the final ceramic replica of the initial polymer template, or the so-called reticulated porous ceramic (RPC). The Schwartzwalder technique can be universally applied and numerous examples of RPCs made from different ceramic materials were described [3,9]. Nevertheless, the manufacturing of cellular cermet structures has not been demonstrated up to now. Consequently, the present work focusses on the preparation of cellular NiO–YSZ foams adapting the Schwartzwalder technique, in the first instance. The obtained ceramic composite foams are subsequently treated with hydrogen-containing gas in order to chemically reduce the NiO phase into metallic Ni. This results in cellular Ni–YSZ cermets in which different ratios of Ni–YSZ were realized. Both, the Ni–YSZ and NiO–YSZ composites, were characterized with respect to their microstructure using scanning electron microscopy (SEM), X-ray diffraction (XRD) with Rietveld analysis, confocal Raman microscopy and porosity analysis. The electric and thermal conductivity and the compressive strength of the obtained NiO–YSZ and Ni–YSZ foams were further investigated and correlated to the microstructure and phase composition of the sample.

## 2. Materials and Methods

### 2.1. Sample Preparation

NiO–YSZ foams were prepared according to the Schwartzwalder, or sponge replication technique. An aqueous dispersion of NiO (Alfa-Aesar GmbH, Karlsruhe, Germany) and YSZ (8 mol.% Y_2_O_3_, Pengda Munich GmbH, Munich, Germany) and polyurethane (PU) foam templates with a linear pore density of 20 ppi (pores per linear inch; SP30P20R, Koepp Schaum GmbH, Oestrich-Winkel, Germany) were used. The weight ratio between NiO and YSZ was adjusted to 1:4, 1:1 and 4:1 (wt./wt.), respectively. The exact procedure for the dispersion preparation and the foam manufacturing was adapted from a previous work [10].

The NiO–YSZ dispersion was prepared by adding 100 g of the NiO–YSZ oxide mixture and 0.88 g ethanolammonium citrate deflocculant (Dolapix CE64, Zschimmer and Schwarz Chemie GmbH, Lahnstein, Germany) to 21.7 mL distilled water. Subsequently, the mixture was homogenized in a planetary centrifugal mixer (2000 rpm for 15 min; THINKY Mixer ARE-250, THINKY Corp., Tokyo, Japan). Afterward, 1.30 g polyvinylalcohol binder (Optapix PA 4G, Zschimmer and Schwarz Chemie GmbH) and 0.09 g polyalkylene glycolether defoamer (Contraspum K1012, Zschimmer and Schwarz Chemie GmbH) were added to the dispersion followed by a second mixing step. The resulting dispersion had a solid content of 80.6 wt.% and was directly applied onto cubic PU templates (20 mm × 20 mm × 20 mm). Each template was completely immersed into the NiO–YSZ dispersion, withdrawn and subsequently freed from the excess amount by manual squeezing. The amount of dispersion applied on each foam was controlled by weighing and was 2.7 ± 0.3 g for all sample series. After drying under ambient conditions the PU template was removed thermally in a circulating air furnace (KU 40/04/A, THERMCONCEPT Dr. Fischer GmbH, Bremen, Germany; 110 °C/2 h, 250 °C/3 h, 400 °C/3 h, heating/cooling rate 1 K∙min^−1^). Afterward, the samples were densified for 3 h in air (heating/cooling rate of 3 K∙min^−1^) in a sintering furnace (HTL 4/18, THERMCONCEPT Dr. Fischer GmbH, Bremen, Germany). The sintering temperature was varied in four steps between 1300 °C and 1600 °C. For each temperature level a sample series of 20 specimens was prepared. The chemical reduction of the NiO to Ni was performed for 10 specimens per sample series at 850 °C in Ar/2 vol.% H_2_ atmosphere in a tube furnace (HTRH 70–600/1800, Carbolite-Gero GmbH and Co. KG, Neuhausen, Germany). The dwelling time at 850 °C was calculated from the total NiO amount in the samples assuming a conversion according to NiO + H_2_ → Ni + H_2_O, a molar volume of 24.5 L∙mol^−1^ and a flow rate of 20 L∙h^−1^ as well as the H_2_ concentration of 2 vol.% in the gas atmosphere. For one sample series the sintering process and the NiO reduction were combined by sintering under Ar/2 vol.% H_2_ atmosphere in a tube furnace operated at 1400 °C, for comparison.

Larger samples for thermal conductivity measurements were prepared by coating PU templates with a dimension of 50 mm × 50 mm × 20 mm. The excess of slurry was extruded with a roller press, template removal, sintering and reduction were performed as described above.

### 2.2. Sample Characterization

The total porosity of the foams (V_pores_/V_foam_) was calculated from the geometric foam density, which is the foam mass m_f_ divided by the geometric foam volume V_f_, and the skeletal density of the strut material. The skeletal density of the NiO–YSZ strut material was calculated by the rule of mixture. The strut porosity (V_strut pores_/V_struts_) was calculated from the dry, buoyant and water-filled weight of the foams as determined by the water immersion/Archimedes’ method according to the DIN EN 623-2:1993-11 standard [11]. Note: The total porosity refers to the geometrical volume of the foam including the foam cells. The strut porosity, on the contrary, refers to the strut volume, which excludes the foam cells. Both numbers must be interpreted separately from each other as their scale basis is different.

The compressive strength was determined with a TIRAtest 2825 universal testing machine (150 mm circular loading plates, crosshead speed of 1 mm∙min^−1^; TIRA GmbH, Schalkau, Germany). A cardboard piece with 1 mm thickness was placed between the foam and loading plates to ensure a more homogeneous load on the sample. The maximum force was extracted from the obtained data and was used for the calculation of the compressive strength. The results of 10 specimens were evaluated assuming a two-parameter Weibull distribution and the average compressive strength σ_cf_ (Weibull scale parameter) together with the modulus m (Weibull shape parameter) as a measure of the Weibull distribution’s width were calculated [12].

For microstructure analysis, selected samples were embedded in epoxy resin, grinded with SiC paper and finally polished with 3 µm and 1 µm diamond dispersions, respectively. These cross-section microsections were then characterized by scanning electron microscopy using a XL30 ESEM-FEG microscope (FEI/Philips, Hillsboro, OR, USA) equipped with a secondary electron (SE) and backscattered electron (BSE) detector. On selected spots, the elemental composition of the Ni(O)-YSZ strut material was analyzed by energy-dispersive X-ray spectroscopy (EDS, EDAX-AMETEK GmbH, Weiterstadt, Germany). Furthermore, selected specimens were investigated by confocal Raman microscopy (Alpha 300R, WITec GmbH, Ulm, Germany). An area scan was performed collecting 25 Raman spectra per µm^2^. The basis Raman spectra of NiO and ZrO_2_ were extracted from the data and fitted to each of the measured Raman spectra (basis analysis) using the WITec Project 2.10 software (WITec GmbH, Ulm, Germany). This procedure allowed a phase mapping presented as a color-coded image showing the phase distribution in the sample.

The quantitative phase composition of the ball-milled NiO–YSZ and Ni-YSZ strut material was determined by powder X-ray diffraction analysis (PANalytical X’Pert Pro Bragg-Brentano diffractometer, Cu-Kα_1_/α_2_ radiation) in θ/θ reflection geometry and a 2θ range from 10° to 100°. The diffraction patterns were analyzed by the Rietveld technique using the Topas Academic 5 package (Coelho Software, Brisbane, Australia) [13].

The thermal conductivity of the NiO–YSZ and Ni–YSZ foams was investigated by the transient plane source (TPS) technique with a TPS 2500 S device (Hotdisk SE, Gothenburg, Sweden). A hotdisk sensor with 9.908 mm in diameter was placed in between two of the rectangular foam samples with previously sanded surfaces and a heating power of 200 mW for a 10 s measurement was used [14]. The thermal conductivity was calculated from the sensor temperature change [15].

Electric conductivity measurements were performed on selected epoxy-embedded specimens, which were grinded and polished from both sides. The polished surfaces were metallized with a thin layer of copper (~1 µm) using a sputter-coater (Metall-Bedampfungsanlage Type B 30.2, VEB Hochvakuum, Dresden, GDR). Electric conductivity measurements were performed by the four-wire technique using planar copper electrodes. A constant current of 50 mA and two multimeters (VC 840, Voltcraft, Hirschau, Germany) were used. From the voltage drop and the sample dimensions, the specific electric conductivity was calculated considering the sample thickness as well as the foam’s cross-sectional area including the porosity. The electric contact between the sample and the electrodes was improved by clamping the entire setup. A screw was tightened using a torque wrench with 15 Nm to a compressive strength of approximately 55 MPa. Each of the composite foams was measured eight times by rotating the sample by 90° and flipping the upper and the lower side. This allowed the elimination of the variations in electric contacting between the copper electrodes and the foam surface.

### 2.3. Sample Naming Scheme

All sample series were named according to the following labelling scheme: **Z***xx***N***yy*-*zzzz*(r/sH_2_) whereas Z*xx* and N*yy* stand for the weight fraction of YSZ and NiO in the dispersion used for the sample preparation and *zzzz* is the sintering temperature. The indices r and sH_2_ refer to reduced samples, or samples sintered in Ar/H_2_ atmosphere, respectively.

## 3. Results and Discussion

### 3.1. Sample Preparation

For all NiO–YSZ dispersions prepared, the flow behavior was sufficient for the manufacturing of cellular ceramics following the Schwartzwalder process. After sintering, intensely green-colored foams were obtained (Figure 1), whereas the color intensified with increasing NiO content. The green color is typical for NiO and the presence of Ni^2+^ cations in an octahedral oxide ligand field. Therefore, no structural changes to the NiO phase by interaction with YSZ was assumed. After treatment with hydrogen-containing gas (98 vol.% Ar, 2 vol.% H_2_), the color is lost and the specimens show a grey shade corresponding to a metallic Ni phase.

The overall mechanical stability of the obtained cellular structures is low and decreases with increasing NiO content. Consequently, all handling steps were performed with great care in order to avoid significant damage to the samples.

### 3.2. Sintering-Induced Shrinkage and Porosity Evolution

The volumetric shrinkage of the NiO–YSZ composite foams increases linearly from 33% after sintering at 1300 °C to 37% for specimens treated at 1600 °C (Z50N50 series; Table 1, Figure 2a). In addition, a clear correlation of the shrinkage behavior to the YSZ weight fraction in the material is found: The shrinkage increases linearly from 25 vol.% for the Z20N80-1400 samples to 50 vol.% for the Z80N20-1400 specimens, respectively. This indicates a significantly higher sintering activity of YSZ compared to the NiO powder, which has also been observed during the densification of NiO–YSZ SOFC anode materials, see references [16,17]. Conversely, the sintering of the YSZ powder should be considered as constrained in the composite foams; the low sintering activity and shrinkage of the NiO limits the compaction of the YSZ phase (Figure 3). For the samples sintered directly in a reducing atmosphere an increased shrinkage of 39 vol.% is observed, which is significantly larger compared to the NiO–YSZ composites sintered in air at the same temperature (34 vol.%). This is a direct consequence of the volume decrease during the NiO → Ni conversion, which occurs at lower temperature and before the sintering of the YSZ powder starts. Consequently, the sintering of the YSZ phase is constrained to a lower extent resulting in a more pronounced shrinkage of the respective composite foams (Figure 3). Furthermore, no additional shrinkage is observed during the hydrogen treatment of the already densified NiO–YSZ composite foams. This is a result of the reduction temperature of 850 °C, which is not sufficient for a further sintering of the material.

The total porosity of the NiO–YSZ and of the Ni–YSZ composite foams is predominantly affected by the Ni(O)-YSZ ratio in the strut material. Consequently, the total porosity is 90.1 vol.% for the Z80N20-1400 samples, 93.0 vol.% for the Z50N50-1400 series and 94.8 vol.% for the Z20N80-1400 foams. These data directly reflect the lower sintering activity of NiO compared to the YSZ powder. The influence of the sintering temperature on the total porosity is less pronounced, and it varies only by ±0.4 vol.% for the Z50N50-*zzzz* sample series. Nevertheless, a slight porosity decrease with increasing sintering temperature is observed. After the hydrogen treatment, the total porosity increases to 91.5 vol.% (Z80N20-1400r), 94.2 vol.% (Z50N50-1400r), 93.9 vol.% (Z50N50-1400sH_2_) and 96.1 vol.% (Z20N80-1400r), respectively. The course of the total porosity values is in good agreement with the tendency in the volumetric shrinkage of the respective samples. The cell size of the foams investigated within this work ranges between 2 mm and 3 mm being in accord to the 20 ppi polymer templates used in sample preparation.

Regarding the porosity inside the ceramic strut material of the NiO–YSZ composite foams (Z50N50 series), a tendency towards lower values is observed with increasing sintering temperature (Table 1, Figure 2b). This is in good accord to the shrinkage behavior and the course of the total porosity of the respective sample series. The specimens treated at 1400 °C show the lowest strut porosity with 34 vol.%—in analogy to results obtained for the sintering of NiO–YSZ SOFC anode materials [18]. However, for the Z50N50-14000 specimens the standard deviation for the strut porosity as well as the shrinkage is higher compared to the other sample series. Therefore, the porosity data for the Z50N50-1400 samples has to be critically scrutinized.

After the hydrogen treatment of the NiO–YSZ composites the strut porosity in the resulting Ni–YSZ foams increases significantly by 12 vol.% (Z50N50 series). The course of the strut porosity is in analogy to the results of the NiO–YSZ samples (Table 1). The drastic increase of the strut porosity during the reduction process can be attributed to the volume decrease during the conversion of NiO (ρ = 6.72 g∙cm^−3^) to Ni (ρ = 8.91 g∙cm^−3^), which represents a volume reduction of 33 vol.% for the Ni-phase. For the specimens which were sintered first and reduced subsequently, the formation of voids between the YSZ matrix and the Ni grains is observed in electron micrographs (Figure 3). This is the result of the above mentioned volume decrease for the Ni-phase and the reduction temperature of 850 °C, which is too low for an ongoing sintering of the YSZ matrix.

If the sintering is performed directly in an atmosphere of Ar/H_2_, the strut porosity is significantly lower with 37 vol.% for the Z50N50-1400sH_2_ specimens compared to 46 vol.% for the Z50N50-1400r series, which was sintered and reduced in two subsequent steps. If the heat treatment is performed in hydrogen-containing atmosphere, the onset of the NiO reduction is at significantly lower temperature (approximately 400 °C for NiO–YSZ composites, reference [19]) compared to the densification processes in the YSZ matrix beginning not below 1000 °C [20]. Consequently, the reduction is finished before the YSZ densification begins and the volumetric shrinkage arising from the NiO → Ni conversion can be compensated by an increased densification of the YSZ matrix. This results in the higher volumetric shrinkage described above as well as a lower strut porosity in these samples.

### 3.3. Microstructure

By powder XRD analysis, cubic YSZ and rhombohedral NiO were identified as only phases in all NiO–YSZ composite foams (Table 1, Figure 4a). The weight fractions of rhombohedral NiO and cubic YSZ are 50 wt.% each in the Z50N50 samples as determined by XRD and Rietveld analyses. No change in the composition and the crystal structure of the respective phases with increasing sintering temperature is observed. For the hydrogen-treated specimens of the Z*xx*N*yy*-*zzzz*r series (see last four samples in Table 1), cubic YSZ, face-centered cubic Ni and rhombohedral NiO were identified as constituents (Figure 4b). However, except for the Z50N50-1300r sample, the NiO concentration is very low and does not exceed 2 wt.%. Consequently, the reduction procedure can be considered as almost quantitative.

In BSE micrographs of NiO–YSZ as well as Ni–YSZ cellular composites the porous structure of the strut material becomes apparent (Figure 5a,b). In addition, a microstructure consisting of two (NiO–YSZ) or three phases (Ni–YSZ) is observed with respect to the grey levels in the recorded micrographs. For the NiO–YSZ samples, these phases can be assigned to NiO and YSZ, respectively, which are homogeneously distributed in the microstructure forming grains with 2 µm to 10 µm in size (Figure 5a). For the reduced samples, darker areas, surrounded by a brighter halo and embedded into a light grey matrix, are observed (Figure 5b,c). The thickness of the halo is 0.2 µm to 0.8 µm. By EDS spectroscopy (Figure 5d), these phases have been identified as a Ni-O compound (dark grey area), pure Ni (bright halo) and a Y/Zr-O compound (light grey area). Regarding the area of the respective phases in the micrographs of the reduced samples, the amount of Ni-O exceeds the fraction of Ni considerably.

A further investigation of the Ni–YSZ composites by confocal Raman microscopy confirms the presence of NiO grains which are surrounded by a Raman-inactive metal phase (Ni) and are embedded into a YSZ matrix (Figure 6). These results are in distinct opposition to the XRD investigations, which suggested an almost quantitative reduction of NiO to Ni.

An explanation for the ternary microstructure of the hydrogen-treated samples could be the formation of amorphous Ni-O intermediates during the interaction with hydrogen. These phases are then hidden for the XRD investigations. However, none of the collected XRD patterns show a pronounced broad maximum in the angular range between 20° and 40° usually observed when a significant amount of amorphous matter is present in the sample. Another possibility is the formation of a porous Ni phase containing traces of oxides, which is surrounded by dense Ni. The porous Ni then appears as darker regions in BSE micrographs surrounded by a brighter halo (dense Ni). This would be in agreement with the XRD results and also in accord to the grey color of the samples after reduction. In summary, these effects need further investigation and clarification. Nevertheless, the successful formation of a nickel phase in the microstructure of the foam struts has been demonstrated.

### 3.4. Compressive Strength

Both the NiO–YSZ and Ni–YSZ composite foams were characterized with respect to their compressive strength, the results are shown in Figure 7a. The overall strength is very low with values in the range between 0.12 MPa and 0.19 MPa for NiO–YSZ and between 0.05 MPa and 0.39 MPa for the Ni–YSZ foams (Table 2). The variation of the individual strength measurements within a sample series, as represented by the Weibull modulus m, is high with m ranging between 1.7 and 4.9, respectively. The low strength is a consequence of the high total porosity of the foams on the one hand, and the porous microstructure of the strut material on the other. Consequently, a clear correlation of the strength to the strut porosity of the respective foams is observed for the NiO–YSZ as well as the Ni–YSZ samples (Figure 7a). Thus, the compressive strength is indirectly correlated to the process parameters “sintering temperature” and “Ni(O)-YSZ ratio”.

The correlation of the compressive strength to the total porosity of the NiO–YSZ and Ni–YSZ cellular composites has been modelled with the Gibson–Ashby relation for brittle, cellular structures (Equation (1), Reference [21])
(1)$$\sigma_{cf}=C{\left(\rho_{rel}\right)}^{n}\sigma_{fs}$$
with *σ_cf_* being the average compressive strength of the cellular material, *ρ_rel_* its relative density (derived from the total porosity P as *ρ_rel_* = 1−P) and *σ_fs_* as bending strength of the bulk strut material. The parameter C is a constant of typically 0.65 for brittle cellular structures. The density exponent n is correlated to the defect concentration within the strut material; in the original Gibson–Ashby relation a value of n = 1.5 is suggested [21]. However, for brittle cellular ceramics the density exponent is usually larger with values between 1.5 and 3, see references [22,23]. As the crushing behavior of both the NiO–YSZ composites and the Ni–YSZ cermets was typically of brittle nature without significant plastic deformation, the Gibson–Ashby model for brittle ceramic foams has been selected.

The modelling of the strength data obtained from the Ni–YSZ cermets was performed using the linearized Gibson–Ashby relation (Equation (2)) and a double-logarithmic compressive strength-relative density plot. This approach allows a simple linear regression analysis [24].

(2)$$ln\left(\sigma_{cf}\right)=ln\left(C\right)+n\left(\rho_{rel}\right)+ln\left(\sigma_{fs}\right)$$

For parameter C the literature value of 0.65 was used and the bending strength *σ_fs_* was set to 245 MPa in accord to literature data obtained for dense Ni–YSZ cermets [25]. The density exponent n was refined as free parameter in order to fit the experimental data. A good approximation was obtained for a density exponent n of 2.4 (Figure 7b), which is in good agreement to previous studies on the strength of cellular ceramics [24,26].

The strength data of the NiO–YSZ samples was modelled on a similar procedure; here, the parameters C and n were fixed to 0.65 and 2.4, respectively, and the bending strength *σ_fs_* was refined. For a *σ_fs_* of 135 MPa for the bulk NiO–YSZ material, which is in agreement to literature data, see reference [27], a good approximation of the experimental strength of the NiO–YSZ foams was obtained (Figure 7b). The Gibson–Ashby approach allows the comparison of the strength of NiO–YSZ and Ni–YSZ cellular composites separated from the severe influence of the sample porosity.

In summary, the compressive strength increases after the reduction of NiO to Ni amounts to 70%, on average, for the same sample porosity. In comparison, for Ni(O)-YSZ SOFC anode materials an increase in flexural strength by approximately 17% after reduction is reported [25]. However, these data do not consider the increase in sample porosity due to the NiO → Ni conversion, but show at least the same general tendency.

### 3.5. Thermal and Electric Conductivity

The room-temperature specific electric conductivity has been investigated for the Z*xx*N*yy*-1400r/sH_2_ sample series as a function of the Ni fraction, which was 11 vol.% (Z80N20-1400r), 36/34 vol.% (Z50N50-1400r/sH_2_) and 67 vol.% (Z20N80-1400r), respectively. The Z80N20-1400r specimen showed no measurable electric conductivity (Table 2, Figure 8). For this sample, the amount of metal in the microstructure is too low in order to form a percolating network and distinct conduction pathways (Figure 9). The room temperature ionic conductivity of YSZ is too low and only relevant at higher temperature (≈ 80 S∙m^−1^ at 1000 °C, reference [28]). For the Ni–YSZ cermet system, a percolation threshold of Φ_p_ = 30 vol.% has been observed for compact samples [29], and was confirmed by theoretical investigations (percolation theory: Φ_p_ = 32 vol.%) [30]. This is in good agreement with the electric conductivity of 2.0∙10^3^ S∙m^−1^ measured for the Z50N50-1400r composite foam, in which the struts contain 36 vol.% Ni. Accordingly, the electric conductivity increases significantly to 18.8∙10^3^ S∙m^−1^ for the Z20N80-1400r samples containing 67 vol.% Ni in the strut material. This effect can be explained by an improved contacting between the increased number of Ni grains in the microstructure, which increases the number of possible conduction pathways (Figure 9).

The sample Z50N50-1400sH_2_, which was sintered directly in hydrogen-containing atmosphere, represents an exceptional position. The electric conductivity is 4.2 × 10^3^ S∙m^−1^ and more than doubled compared to the Z50N50-1400r specimen, which was reduced subsequent to the sintering process in air. For both composite foams the Ni content is similar (36 vol.% and 34 vol.%) and clearly above the percolation threshold. Consequently, the improved electric conductivity of the Z50N50-1400sH_2_ composite is a consequence of the higher shrinkage and increased densification for these samples. By this, the Ni grains are compressed to each other, which improves the contacting between them and increases the number of conduction pathways (Figure 9).

In order to evaluate the electric conductivity data measured for the cellular Ni–YSZ composites against literature results for Ni–YSZ SOFC anode materials, the influence of the sample’s (cellular) porosity has to be considered. For this purpose, a simple model derived by Gibson and Ashby has been applied for the estimation of the bulk electric conductivity κ_B_ (Equation (3), Reference [1])
(3)$$\kappa_{B}=3\kappa_{cf}/\left[\left(1-P\right)+2{\left(1-P\right)}^{3/2}\right]$$
with κ_cf_ as the electric conductivity of the composite foam and the total porosity P. From this relation a bulk electric conductivity of 7 × 10^4^ S∙m^−1^ (Z50N50-1400r), 14 × 10^4^ S∙m^−1^ (Z50N50-1400sH_2_) and 104 × 10^4^ S∙m^−1^ (Z20N80-1400r), respectively, has been calculated. The κ_B_ values for the Z50N50 samples are in good agreement to data collected for Ni–YSZ cermets by Dees and coworkers (reference [29]), which were extrapolated to room temperature (Figure 10). The calculated bulk conductivity of the Z20N80-1400r sample is in the regime of typical metallic conductors (>10^6^ S∙m^−1^).

Nevertheless, it should be noted that the numbers obtained for the Ni–YSZ bulk electric conductivity should be interpreted with care. The Gibson–Ashby model in Equation (3) has been initially designed for the evaluation of the electric properties of pure metallic foams, not for the modelling of data collected from cellular ceramic-metal composite structures. Therefore, a homogeneous strut material is assumed and all porosity is considered as cellular voids (e.g., no strut porosity). Both is a good approximation of most of the typical metal foam structures but not the case for the materials investigated within this work. Nevertheless, the numbers give a conservative estimate of the electric conductivity of the (cellular) Ni–YSZ composites prepared within this study.

The thermal conductivity has only been investigated for a selected specimen of the Z50N50-1400 series which was measured before and after the chemical reduction of the NiO with hydrogen. For both samples, the thermal conductivity is almost equal with 0.23 W∙m^−1^∙K^−1^. Regarding the microstructure of the investigated samples, this is not unexpected, despite the better thermal conductivity of Ni (79 W∙m^−1^∙K^−1^) compared to NiO (35 W∙m^−1^∙K^−1^) [31]. However, as discussed before, a significant increase in porosity (total and strut porosity) has been observed after the hydrogen treatment. These additional voids reduce the effective thermal conductivity of the cellular Ni–YSZ composites, regardless the hypothetically benefit from the higher conductivity of the Ni phase.

## 4. Conclusions

The manufacturing of cellular NiO–YSZ ceramic composites with a total porosity exceeding 92.5 vol.% has been successfully demonstrated by following the Schwartzwalder process using aqueous NiO–YSZ dispersions. The obtained foams are characterized by a high strut porosity, which limits the mechanical strength to low values around 0.15 MPa. In summary, this is a consequence of the low sintering activity of NiO compared to YSZ, which results in the formation of the aforementioned strut porosity.

After a hydrogen treatment of the sintered NiO–YSZ foams cellular Ni–YSZ cermet structures can be obtained. These show an again increased strut porosity over the NiO–YSZ samples due to the volume decrease during the chemical NiO → Ni conversion. Nevertheless, their compressive strength is 70% higher compared to NiO–YSZ samples at the same porosity level. The electric conductivity of these Ni–YSZ cermets is promising with values between 2 × 10^3^ S∙m^−1^ and 19 × 10^3^ S∙m^−1^ for the cellular structure in its as prepared form. If the sintering of the NiO–YSZ green bodies is performed directly in hydrogen-containing atmosphere, the strut porosity is reduced significantly resulting in an increase of the electric conductivity by 110% at the same Ni volume fraction. If the influence of the sample porosity is modelled with appropriate porosity ↔ conductivity models, a good fit to data obtained for conventional Ni–YSZ SOFC anode materials is found.

## Figures and Tables

**Figure 1 materials-13-02437-f001:**
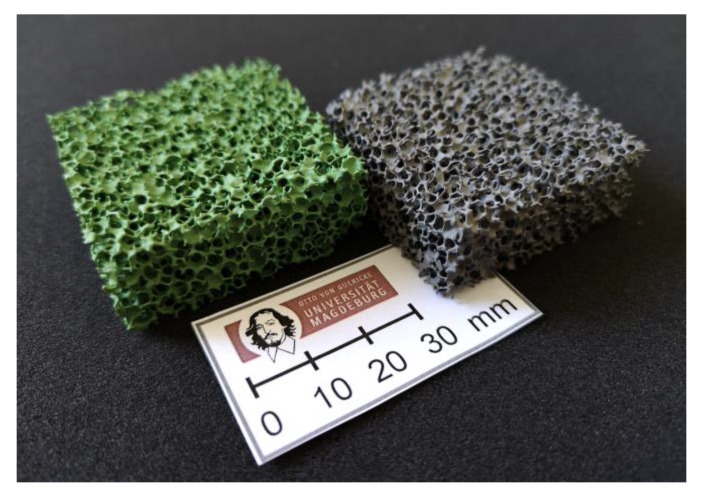
Specimens of NiO–YSZ (left) and Ni–YSZ cellular composites (right). The ratio of NiO and YSZ is 1:1 by weight. Before reduction, the foams possess the typical green color of NiO, which transforms to grey after reduction to metallic Ni.

**Figure 2 materials-13-02437-f002:**
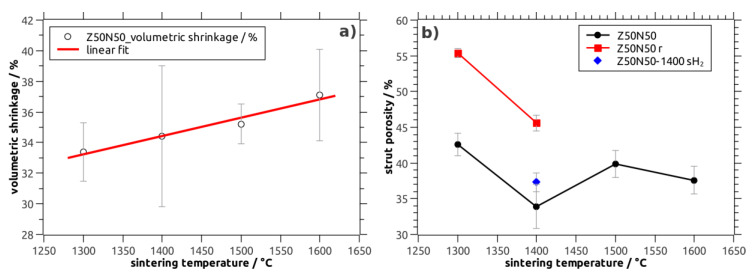
(**a**) Volumetric shrinkage during sintering of NiO–YSZ composite foams (Z50N50 series) as a function of the sintering temperature; (**b**) strut porosity in NiO–YSZ and Ni–YSZ cellular composites as a function of the sintering temperature.

**Figure 3 materials-13-02437-f003:**
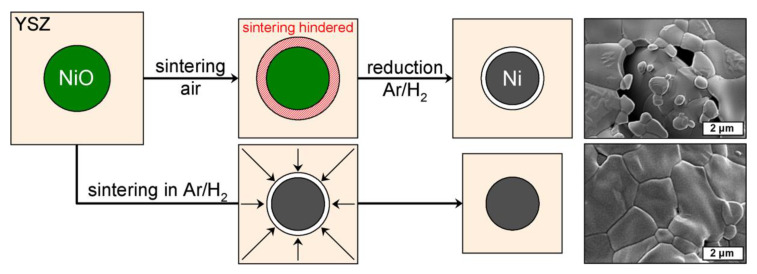
Schematic representation of the sintering of NiO–YSZ composites: *Top:* Subsequent sintering and reduction-the densification of YSZ is hindered by the NiO, which is less active towards sintering → voids are formed after reduction to Ni; *bottom:* combined sintering and reduction in Ar/H_2_ atmosphere-NiO is reduced to Ni before sintering of YSZ starts → higher shrinkage of the composite, less formation of voids.

**Figure 4 materials-13-02437-f004:**
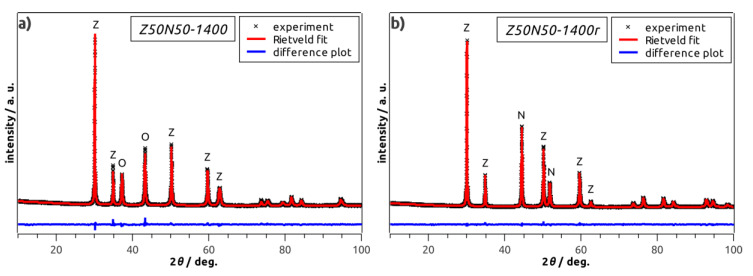
Powder XRD patterns of selected Z50N50-1400 specimens before (**a**)*,* and after the hydrogen treatment (**b**). The letters Z, N, O correspond to the main reflections of YSZ, Ni and NiO, respectively.

**Figure 5 materials-13-02437-f005:**
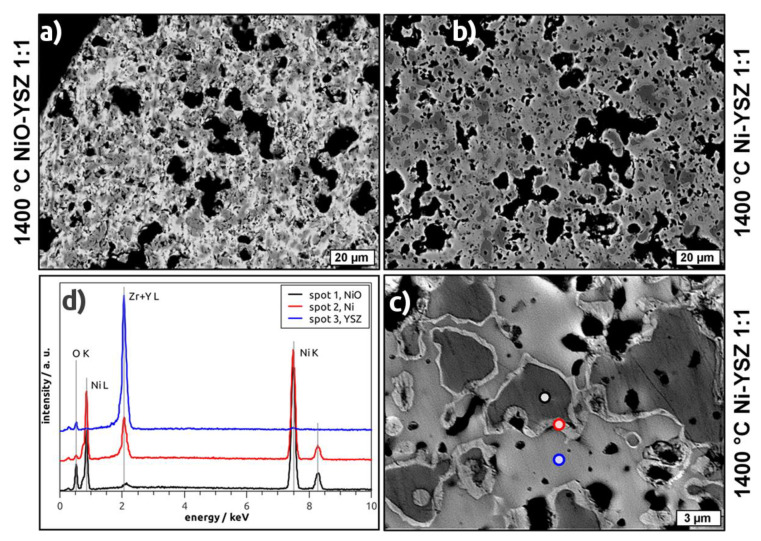
Backscattered electron micrographs of selected samples containing equal weight fractions of NiO and YSZ and sintered at 1400 °C before (**a**) and after reduction with H_2_ (**b**,**c**), black regions correspond to pores inside the material; (**d**) energy-dispersive X-ray spectroscopy (EDS) spectra recorded on characteristic spots show the element distribution in the respective Ni/NiO/YSZ phases being present in the hydrogen-treated samples.

**Figure 6 materials-13-02437-f006:**
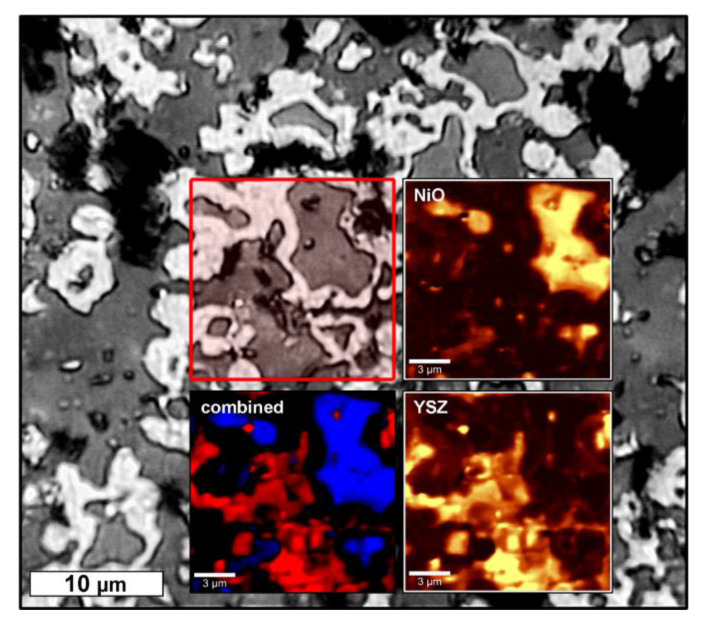
Confocal Raman-microscopic analysis of a crossection–microsection of the Z50N50-1400r sample. The greyscale image is an optical micrograph and the red-framed area was analyzed by Raman spectroscopy. The distribution of the NiO and YSZ phases in the scanned area is shown as inset. The Ni phase is represented as black areas in the color-coded combination of the NiO (blue) and YSZ (red) distribution.

**Figure 7 materials-13-02437-f007:**
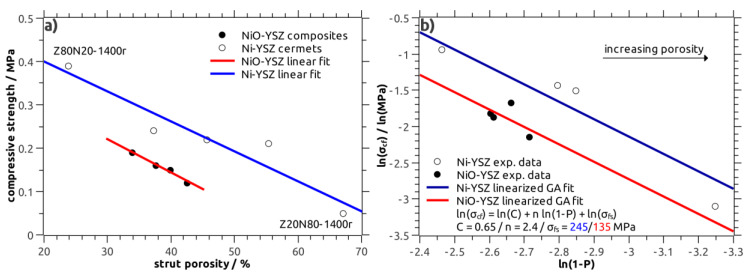
(**a**) Correlation between the compressive strength and the strut porosity in NiO–YSZ composites (filled symbols, red line) and Ni–YSZ cermets (open symbols, blue line); (**b**) Gibson–Ashby (GA) fit of the strength data obtained from NiO–YSZ and Ni–YSZ composites.

**Figure 8 materials-13-02437-f008:**
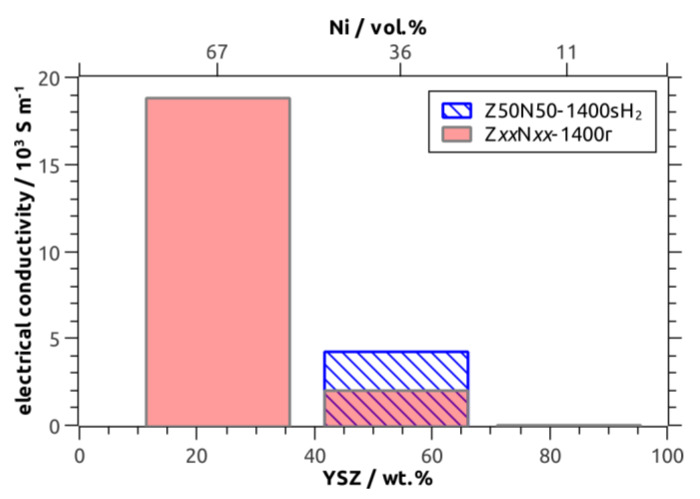
Electric conductivity of Ni–YSZ cermets as a function of the YSZ weight fraction/Ni volume fraction according to XRD with Rietveld analysis.

**Figure 9 materials-13-02437-f009:**
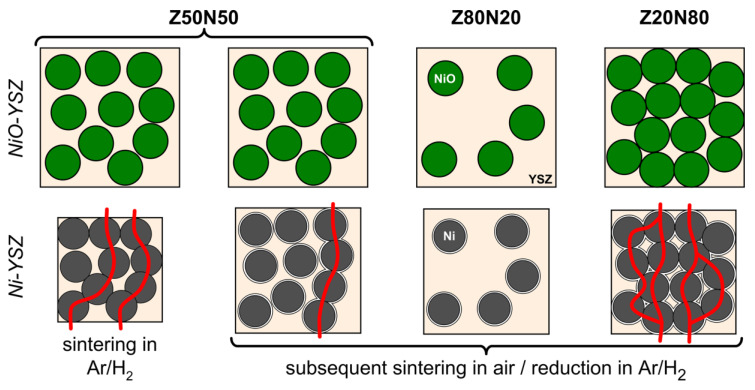
Schematic representation of the evolution of conduction pathways (red) in the microstructure of Ni–YSZ cermets as a function of the NiO content and the sintering conditions.

**Figure 10 materials-13-02437-f010:**
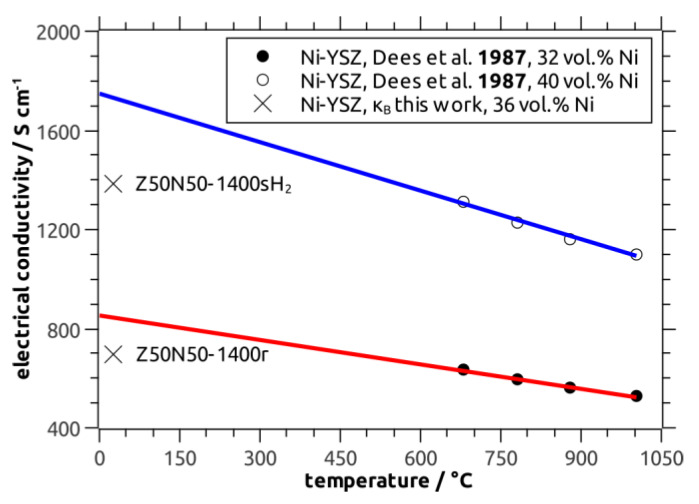
Bulk electric conductivity of Z50N50 Ni–YSZ cermets (36 vol.% Ni) calculated using Equation (3) (X) compared to literature data measured for Ni–YSZ with 32 vol.% Ni (closed symbols, red line) and 40 vol.% Ni (open circles, blue line). Note: the literature data were extrapolated to room temperature applying a linear regression.

**Table 1 materials-13-02437-t001:** Volumetric shrinkage, total porosity (P_total_) and strut porosity (P_strut_) as well as phase composition of NiO–YSZ and Ni–YSZ composite foams. The phases identified by X-ray diffraction (XRD) are cubic YSZ, rhombohedral NiO and cubic Ni.

Sample	Shrinkage/vol.%	P_total_ ^a^/vol.%	P_strut_ ^b^/vol.%	c-YSZ/wt.%	r-NiO/wt.%	Ni/wt.%	Ni/vol.%
Z50N50-1300	33.4 ± 1.9	93.4 ± 1.1	42.6 ± 1.6	51.3	48.7	0	0
Z50N50-1400	34.4 ± 4.6	93.0 ± 0.8	33.9 ± 3.1	49.5	50.5	0	0
Z50N50-1500	35.2 ± 1.3	92.7 ± 0.6	39.9 ± 1.9	50.1	49.9	0	0
Z50N50-1600	37.1 ± 3.0	92.6 ± 0.1	37.6 ± 1.9	49.8	50.2	0	0
Z50N50-1400sH_2_	39.1 ± 1.3	93.9 ± 0.2	37.3 ± 1.3	55.9	0	44.1	34
Z20N80-1400	25.1 ± 1.7	94.8 ± 0.4	50.3 ± 4.0	−/−	−/−	−/−	−/−
Z80N20-1400	50.0 ± 2.3	90.1 ± 0.3	26.8 ± 2.3	−/−	−/−	−/−	−/−
Z50N50-1300r	−/−	94.5 ± 0.1	55.4 ± 0.6	52.8	13.1	34.1	26
Z50N50-1400r	−/−	94.2 ± 0.5	45.6 ± 1.1	53.9	0.6	45.5	36
Z20N80-1400r	−/−	96.1 ± 0.5	67.1 ± 1.0	23.4	1.7	74.9	67
Z80N20-1400r	−/−	91.5 ± 0.6	23.8 ± 1.6	83.1	1.3	15.7	11

^a^ scale basis: total foam volume, P_total_ = V_pores_/V_foam_∙100%; ^b^ scale basis: strut volume, P_strut_ = V_strut pores_/V_strut_∙100%.

**Table 2 materials-13-02437-t002:** Compressive strength σ_cf_ (Weibull average of 10 specimens), electric conductivity κ_el_ and thermal conductivity λ_cf_ of NiO–YSZ and Ni–YSZ composite foams.

Sample	σ_cf_/MPa	Weibull Modulus (m)	κ_cf_/10^3^ S∙m^−1^	λ_cf_/W∙m^−1^∙K^−1^
Z50N50-1300	0.12	4.0	−/−	−/−
Z50N50-1400	0.19	4.0	−/−	0.232 ± 0.005
Z50N50-1500	0.15	4.6	−/−	−/−
Z50N50-1600	0.16	3.3	−/−	−/−
Z50N50-1400sH_2_	0.24	5.8	4.2 ± 0.6	−/−
Z50N50-1300r	0.21	2.9	−/−	−/−
Z50N50-1400r	0.22	2.7	2.0 ± 0.1	0.229 ± 0.009
Z20N80-1400r	0.05	1.7	18.8 ± 4.6	−/−
Z80N20-1400r	0.39	4.9	0	−/−

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
