# Peer review of "Cellular Nickel-Yttria/Zirconia (Ni–YSZ) Cermet Foams: Manufacturing, Microstructure and Properties"

_materials, 2020, doi:10.3390/ma13112437_

Round 1

Reviewer 1 Report

Ni-ZrO2-Y2O3  (Ni-YSZ) gained popularity as anode material in solid oxide fuel cells (SOFCs) due to its unique combination of ionic conductivity through the YSZ phase and electronic conductivity through the Ni phase. Thus, the relevance of research in this area is obvious. In this paper, the manufacturing of cellular NiO-YSZ and Ni-YSZ ceramic composites has been sucessfully demonstrated. The NiO-YSZ ceramic foams, as well as the cellular Ni-YSZ cermets prepared therefrom,  were  characterized  with  respect  to  their  microstructure  by  scanning  electron microscopy, confocal Raman microscopy and X-ray diffraction with Rietveld analysis. In addition, the compressive strength, the electric conductivity and the thermal conductivity were determined. Cellular Ni-YSZ cermet structures show increased strut porosity over the NiO-YSZ samples. Their compressive strength is higher compared to NiO-YSZ samples at the same porosity level and the electric conductivity of these Ni-YSZ cermets is promising. Electric conductivity can be adjusted by both, the Ni volume fraction, and the sintering/reduction procedure. The paper is interesting, the results and interpretation seem adequate. I recommend it for publication in present form.

Author Response

Thank you very much for reviewing our paper and the positive feedback.

Reviewer 2 Report

In this manuscript, Betke and his co-workers present a novel synthesis of cellular NiO-yttria-stabilized zirconia (NiO-YSZ) foams adapting the Schwartzwalder technique, and the reduction of NiO-YSZ to cellular Ni-ZrO2-Y2O3 (Ni-YSZ) cermets. Both the NiO-YSZ and Ni-YSZ composites have been characterized using scanning electron microscopy, confocal Raman microscopy and X-ray diffraction. In addition, the compressive strength, the electric conductivity and the thermal conductivity were also determined. The research is carefully done and the manuscript is well-written. Conclusions are based on experimental findings. Unresolved problems concerning the microstructural analysis, viz. NiO content of Ni-YSZ, have also been pointed out. The content of the manuscript fits the scope of the journal. I have found this manuscript interesting and well-organized, and the experimental results well-interpreted. I suggest the acceptance of this manuscript in its present form.

Author Response

(The authors gave the same response as above.)

Reviewer 3 Report

Cermet foams are having so many industrial applications, especially for high temperature and corrosive atmospheres. Hence this publication may be considered for this journal. However, following suggestions or comments may be observed to improve the quality of manuscript and make it more reader friendly:

  1. Key words looks not systematic: it can be like 2 key words on the name of materials involved, 2 on processing or fabrication method and 2 on characterisation techniques will be looks more attractive and get appropriate for Google or other search engines.
  2. Please add error bars to Fig. 2
  3. 3 – add microstructure for sintered in Ar/H2 sample too in order to make it more reader friendly
  4. 2 vol% H2 seems to be too low – is it enought to reduce the NiO?
  5. For each condition – how many samples prepared and/or analysed?
  6. In the microstructure section – it is good if you would add the pore size and its distribution’s statistics and compare the data with different samples
  7. Cermet foams are potential for high temperature applications, so it will be good if you would present the high temperature thermal & electrical conductivity.
  8. Conclusions may be presented in point or bullet wise instead of paragraphs. It is optional suggestion only.

Author Response

Thank you very much for reviewing our paper and the feedback. The following improvements were applied to the manuscript:

1) the list of keywords has been revised and is now better structured

2) figure 2 has error bars now

3) a microstructure image of the foam sintered under Ar/H2 is incluided in figure 3 now

4) the low H2 concentration in the reducing gas atmosphere has been compensated by extending the reaction time. This has been clarified in the experimental part (ll. 93-94).

5) for each series 20 specimens were prepared from which 10 were reduced. Therefore, the porosity and shrinkage measurements were performed on 10 samples per series. The electrical conductivity was measured on one sample per series, the strength on the remaining 9. For the thermal conducxtivity, separate samples were prepared, which were larger in size.

6) no tomographic data is available for the samples within this work, therefore, the analysis of the pore size distribution has not been performed. However, from other studies on cellular ceramics within our group a direct correlation of the cell size to the linear shrinkage of the samples has been identified.

7) The electrical conductivity measurerments were performed on samples, which were embedded into expoxy rersin previously. This is neccessary for a good contacting of the samples by clamping them with high pressure between the electrodes. Without clamping, the contact resistance between sam,ple and electrodes is high and does not allow a reliable measurment of the electric conductivity. The epoxy resin is not stable at elevated temperature, therefore, the conductivity measurements were only performed at room temperature.

Reviewer 4 Report

The authors U. Betke et al. studied NiO- and Ni-YSZ-based foams prepared using Schwartzwalder method.

It seems that experimental work was well done, however a description of the results should be improved.

  1. The authors immerse PU foams with defined dimensions in different dispersions containing various NiO: YSZ ratio. After tha,t the coated PU foams were squeezed in order to remove excess of the dispersions. The authors do not mention in the manuscript how much of the dispersions is left in the PU foams, or with other words, how the authors provide that each sample contain same loading of NiO-YSZ particles, because squeezing was performed manually. So, how a reproducibility was provided and what was a mass of each sample after burn out of polymer.
  2. Are the authors able to explain why the strut density reaches the highest value after sintering at 1400oC, but it decreases with increasing as well as decreasing of the sintering temperature (Fig. 2b).
  3. Since the authors confirmed by XRD that components NiO, Ni and YSZ do not react at the sintering process, why then they performed Rietveld analysis? (and again they confirmed that that the ratio of the components is the same as was in the starting mixture?).
  4. Line 262: The authors write: “…in the microstructure forming domains with…” On the ceramic field the term “domain” is somehow reserved for piezoelectric (magnetic) ceramics where the term domain stands for homogeneous volume of a ceramic where dipoles are oriented in the same direction.
  5. SEM micrographs on the Fig. 3 and 5 b, c show microstructure of the reduced sample, but are quite different. Why?
  6. Line 270: The authors write: “…presence of NiO agglomerates…”. Actually, the term agglomerate is usually used when we describe nano- or micrometer sized powders, and not ceramics.
  7. A reader may miss an estimation of pore size of the prepared foams. Can you include this information, which is crucial information when we describe porous materials.
  8. Line 361: In ceramics there are no particles! but grains, areas, ….
  9. For electrical measurements the authors sputtered Cu electrodes on the surface of their specimens. Since the specimens have pretty large pores (I can estimate from the Fig. 1 that the size of pores can exceed 2 mm), so the sputtering of Cu may go pretty deeply inside the samples? How do the authors evaluate/estimate then a distance between electrodes?
  10. When the authors mention in the manuscript that NiO reduces into Ni (NiO to Ni conversion) they never mention that actually there is always only a partial conversion! (as microstructural analysis shows), so be more précised.
  11. Such description, Fig. 5 a+b, is not common in scientific journals.

Author Response

Thank you for reviewing the paper and your constructive feedback. The manuscript has been altered as follows:

1) The weight of the samplers has been controlled by weighing the coated foams and the ammount of dispersion was adjusted to 2.7 g for all samples with a standard deviation of 0.3 g. The experimental section has been updated accordingly (ll. 83-84)

2) The same finding has been observed in a study of bulk YSZ-NiO composites. However, no explanation is given. Regarding this work, the standard deviation of the strut porosity for the 1400 °C samples is higher compared to the other series, consequently, the has to be interpreted with care. The manuscript has been updated accordingly (error bars in Fig. 2 and ll.215-217)

3) Rietveld analysis was performed in order to prove, that there is no reaction between YSZ and NiO. This is not only limited to the raw phase identification and quantification, but also to microstructural data regarding changes to the crystal structure, e. g. occupation of crystallographic sites, or phase transitions.

4) The terminus domain has been changed to grain

5)  The SEM image in Fig. 3 shows a secondary electron micrograph of the strut surface of a reduced foam and the images in Fig. 5 are backscattered electron micrographs recorded on crosssection-microsections of the same sample

6) The terminus agglomerates has been substituted by grains.

7) no tomographic data is available for the samples within this work, therefore, the analysis of the pore size distribution has not been performed. The cell size has been evaluated qualitatively, an information has been included into the manuscript (ll. 209-211)

8) The terminus particles has been substituted by grains.

9) As stated in the experimental section, the electric conductivity was measured on samples embedded into epoxy resin and subsequently grinded/polished. Therefore, the contact area is even anmd the distance can be exactly measured and a homogeneous copper coating can be applied.

10) The XRD analyses show a quantitative reduction and is a quantitative analysis at least for the crystalline matter. The qiualitative microstructure images show different grey levels and probably an incomplete reduction. However, the differing grey levels can also originate from a porosity in the respective regions. This discrepancy has been thoroughly discussed within the text (ll. 276-296), therefore the authors think, that the terminus addressing the NiO reduction is appropriate.

11) The figures are addressed as Figure 5a and 5b now.

Round 2

Reviewer 4 Report

I can further discus about authors answers, but I will not. The authors mainly followed the proposed suggestions and correct manus

I can further discus about authors answers, but I will not. The authors mainly followed the proposed suggestions and correct manuscript accordingly.